# Paths of Evolution of Progressive Anaplastic Meningiomas: A Clinical and Molecular Pathology Study

**DOI:** 10.3390/jpm13020206

**Published:** 2023-01-25

**Authors:** Rina Di Bonaventura, Liverana Lauretti, Maurizio Martini, Tonia Cenci, Giuliano Di Monaco, Davide Palombi, Giovanni Maria Ceccarelli, Silvia Chiesa, Marco Gessi, Alessia Granitto, Alessio Albanese, Luigi Maria Larocca, Quintino Giorgio D’Alessandris, Roberto Pallini, Alessandro Olivi

**Affiliations:** 1Department of Neurosurgery, Fondazione Policlinico Agostino Gemelli IRCCS, Università Cattolica del Sacro Cuore, 00168 Roma, Italy; 2Department of Pathology, Fondazione Policlinico Agostino Gemelli IRCCS, Università Cattolica del Sacro Cuore, 00168 Roma, Italy; 3Department of Radiation Oncology, Fondazione Policlinico Agostino Gemelli IRCCS, Università Cattolica del Sacro Cuore, 00168 Roma, Italy

**Keywords:** meningioma, anaplastic, progression, EGFRvIII, PD-L1, Sox2

## Abstract

Grade 3 meningiomas are rare malignant tumors that can originate de novo or from the progression of lower grade meningiomas. The molecular bases of anaplasia and progression are poorly known. We aimed to report an institutional series of grade 3 anaplastic meningiomas and to investigate the evolution of molecular profile in progressive cases. Clinical data and pathologic samples were retrospectively collected. VEGF, EGFR, EGFRvIII, PD-L1; and Sox2 expression; *MGMT* methylation status; and *TERT* promoter mutation were assessed in paired meningioma samples collected from the same patient before and after progression using immunohistochemistry and PCR. Young age, de novo cases, origin from grade 2 in progressive cases, good clinical status, and unilateral side, were associated with more favorable outcomes. In ten progressive meningiomas, by comparing molecular profile before and after progression, we identified two subgroups of patients, one defined by Sox2 increase, suggesting a stem-like, mesenchymal phenotype, and another defined by EGFRvIII gain, suggesting a committed progenitor, epithelial phenotype. Interestingly, cases with Sox2 increase had a significantly shortened survival compared to those with EGFRvIII gain. PD-L1 increase at progression was also associated with worse prognosis, portending immune escape. We thus identified the key drivers of meningioma progression, which can be exploited for personalized treatments.

## 1. Introduction

Meningiomas are mostly benign central nervous system tumors, originating from arachnoid cap cells [1]. The existence of aggressive cases has been recognized since the beginning of the 20th century, when Cushing et Eisenhardt reported a meningioma patient surviving only 2.5 years [2]. According to the WHO classification [1], anaplastic meningiomas are sorted as grade 3 (grade III in 2016 and previous WHO classifications [3]). Anaplastic meningiomas represent only 5% of all meningioma tumors but are overt malignancies with a dismal prognosis. A population-based study showed the 5-year survival rate of patients with anaplastic meningiomas of 41.4% in the USA [4] and 68.9%, in Korea [5]. Due to its rarity, there are very few reports in the literature on the progression pattern and the response to the treatment of anaplastic meningioma, and prognostic factors are unclear and controversial. GTR seems to be associated with better survival outcomes, and adjuvant radiotherapy is recommended regardless of the extent of resection. Available chemotherapy has limited efficacy. Anaplastic meningiomas can arise de novo or result from the progression of grade 1 or 2 meningiomas [1,3]. The molecular bases of anaplasia in meningioma are poorly known. Historical studies focused mainly on cytogenetic alterations [6,7]. Recently, using the NGS platform, omics have been investigated in meningiomas [8] with interesting results, which are difficult to translate into the clinical practice. In a previous study that used immunohistochemistry and real-time RT-PCR, we showed that the expression of the stemness marker Sox2 is associated with a worse prognosis independently from the WHO meningioma grade [9]. In particular, we showed that Sox2-expressing grade 1 and 2 meningiomas are associated with an increased risk of progression to anaplastic tumors. However, apart from Sox2, the paths of molecular evolution of anaplastic meningiomas largely remain to be investigated.

Next, in terms of Sox2, in this work we analyzed other markers involved in different molecular pathways, namely: VEGF, the vascular endothelial growth factor, a major regulator of angiogenesis; EGFR, the epidermal growth factor receptor, regulating growth, survival, proliferation, and differentiation in mammalian cells; EGFR variant III (EGFRvIII), the most common extracellular domain mutation of EGFR with ligand-independent constitutive signaling activation, which is a known driver of tumor progression; programmed death ligand-1 (PD-L1), whose pathway controls the induction and maintenance of immune tolerance within the tumor microenvironment through T cell activation, proliferation, and cytotoxic secretion; *MGMT* (*O6-methylguanine-DNA methyltransferase*), a DNA repair enzyme that plays an important role in chemoresistance to alkylating agents; *TERT* (*telomerase reverse transcriptase*), the catalytic subunit of telomerase, whose modulation can be performed through promoter germline and somatic mutations, gene amplifications, structural variants, and epigenetic changes.

The present work focuses on anaplastic meningiomas. More specifically, we used widely available techniques, such as immunohistochemistry and PCR, to assess the molecular paths of the evolution of progressive anaplastic meningiomas.

## 2. Materials and Methods

### 2.1. Patients Enrollment and Collection of Clinical Data

We retrospectively enrolled 41 consecutive patients undergoing neurosurgery for anaplastic meningioma (WHO grade III according to WHO 2016) at the Department of Neurosurgery, A. Gemelli University Hospital, Rome, Italy, between 1999 and 2018 [9]. Baseline demographics; clinical status, according to modified Rankin scale; tumor location (classified as skull base, non-skull base, and intra-ventricular); and the number of neurosurgeries, surgical complications, extent of resection (gross total resection, GTR, i.e., Simpson 1–3, vs. subtotal resection, STR, i.e., Simpson 4–5), and adjuvant treatments were registered. Tumor progression was evaluated using RECIST ver. 1.1 criteria. Overall survival (OS) was calculated from the date of surgery in which a diagnosis of grade III meningioma was established to death from any cause or last follow-up.

This study was approved by the ethics committee of A. Gemelli University Hospital (study ID 3459).

### 2.2. Molecular Analysis

Molecular analysis was performed in a subset of 10 patients suffering from progressive anaplastic meningioma. Formalin-fixed paraffin-embedded paired tumor samples were collected both at the first surgery (pre-anaplastic sample) and at surgery where a diagnosis of anaplastic meningioma was established. In these samples, we evaluated the expression of VEGF, EGFR, EGFRvIII, PD-L1, Sox2, *MGMT* methylation status, and *TERT* promoter mutation. These players were selected among the potentially actionable molecular markers involved in tumor proliferation and aggressiveness, already studied and revealed as relevant in other tumors, and also considering the availability of standard histopathology analysis techniques [10,11,12,13,14,15,16]. The selection of the evaluation technique was decided according to the literature evidence of its value and common availability in daily practice. VEGF expression was evaluated using immunohistochemistry and EGFRvIII expression using RT-PCR and *MGMT* promoter methylation using methylation-specific PCR [17,18,19]. Sox2 expression was evaluated using immunohistochemistry, as already described. A sample was scored as positive where ≥25% cells showed nuclear or nuclear-cytoplasmic expression of Sox2 [9]. EGFR and PD-L1 were assessed immunohistochemically using anti-human EGFR (1:50, clone C1, UCS diagnostics, Morlupo, Italy) and anti-human PD-L1 (1:50; clone 22C3, Dako, Milan, Italy). EGFR expression was scored as 0, 1+, 2+, or 3+ in case of positivity of >25%, 25–50%, 50–75%, or >75% of tumor cells, respectively (Appendix A) [20]. PD-L1 expression was evaluated as absolute percent of positive tumor cells (Tumor Proportion Score, TPS; Appendix A) [21]. For the analysis of *TERT* promoter mutations, the following primers were used for DNA amplification, F: 5′-CAC CCG TCC TGC CCC TTC ACC TT-3′ R: 5′-GGC TTC CCA CGT GCG CAG CAG GA-3′. The annealing temperature was 62 °C. The 230 bp amplified product was purified by adding 2 μL of ExoSap (USB Corporation, Cleveland) at 37 °C for 15 min, and at 80 °C for 15 min. Sequencing was carried out using BigDye Terminator v3.1 cycle sequencing kit (Applied Biosystem, Waltham, MA, USA) in a final volume of 20 μL. Samples were finally processed using ABI Prism 3700 Avant (Applied Biosystem).

### 2.3. Statistical Analysis

Continuous variables were described using mean ± standard deviation (SD) or median and range. Comparison of continuous variables between groups was performed using the Mann–Whitney U test. Categorical variables were compared using chi-square statistic, applying the Fisher’s exact test when appropriate. Survival curves were plotted using Kaplan–Meier method and analyzed using log-rank test. Multivariate analysis for survival was performed using Cox proportional hazards model, adjusting for the following variables, age (> or <70 years), phenotype (progressive or de novo), postoperative performance status (mRS ≤ 3 or >3), side (bilateral or unilateral), extent of resection (GTR vs. STR). Analyses were performed using software GraphPad Prism 7 (GraphPad Software, la Jolla, CA, USA), *p* < 0.05 was considered as significant.

## 3. Results

### 3.1. Baseline Characteristics

We enrolled 41 patients operated for grade III meningioma. Seventeen cases were primary (de novo) meningioma, while 24 cases were progressive from lower grade meningioma. The latter tumors originated from a grade I meningioma in 38.1% of cases, and from a grade II meningioma in 61.9% of cases. The average time of transformation from a previous meningioma to grade III meningioma was 30 months (range, 13–74 months), with no significant differences between the ones coming from grade I and grade II. The average follow-up time was 2.1 years (range 1–3.8 years). The clinical features of patients are detailed in Table 1. No significant differences were detected between de novo and progressive meningioma regarding gender, age, tumor location, and pre- and post-operative clinical status. In progressive cases, no significant trend to a more frequent skull base location was noticed.

### 3.2. Follow-Up

Surgical and follow-up features of anaplastic meningiomas are shown in Table 2 and Appendix A. Most patients received a single surgery after the diagnosis of anaplastic meningioma, with no significant differences between de novo and progressive meningioma. In the pre-anaplastic history of progressive cases, 52.4% of patients underwent a single operation, with no significant differences between cases originating from grade I and from grade II (Appendix A).

Gross total resection was obtained more frequently in de novo vs. progressive cases (*p* = 0.0237, Fisher’s exact test). The average complication rate was 34.3%: this figure is in line with literature data, which report a 40% average complication rate [22]. All but three patients underwent adjuvant radiotherapy. The reason for these latter cases was the poor postoperative performance status. Adjuvant chemotherapy was performed in five progressive patients and included fotemustine, somatostatin analogues, rapalogs, and bevacizumab.

### 3.3. Clinical Prognosticators

Median OS since grade III diagnosis was 2.7 years. Clinical prognosticators are shown in Figure 1. Young age, de novo cases, origin from grade II in progressive cases, good pre- and post-operative clinical status, and unilateral side were significant factors for good outcome. Instead, gender, tumor location, and extent of surgical resection were not associated with prognosis. On multivariate analysis (Table 3), no independent prognosticators of OS were detected. The meningioma phenotype (de novo vs. progressive) and age (≤70 vs. >70) showed a trend of significance.

### 3.4. Molecular Profiling of Progressive Anaplastic Meningiomas

In a subset of 10 patients suffering from progressive anaplastic meningioma, we assessed the molecular profile on paired tumor samples collected both at first (pre-anaplastic) surgery and at surgery in which the diagnosis of grade III (anaplastic) meningioma was established. In these tumors, we assessed the expression of VEGF, EGFR, PD-L1, Sox2, EGFRvIII, *TERT* promoter mutation, and *MGMT* promoter methylation. Results of these analyses are presented in Figure 2 and Figure 3. In detail, VEGF was overexpressed in all cases in pre-anaplastic tumors and in all but one anaplastic sample. EGFR was overexpressed in all cases in a pre-anaplastic sample; however, at progression, a reduction in immunohistochemical staining was observed in 60% of cases. Conversely, EGFRvIIII was positive in 60% of pre-anaplastic samples and in 90% of anaplastic tumors, with three cases showing EGFRvIII gain. A wildtype *TERT* promoter was observed in all samples both before and after progression. Similarly, *MGMT* promoter was unmethylated in all but one case. Interestingly, PD-L1 was poorly expressed in pre-anaplastic samples, while expression levels increased in 40% of cases at anaplastic progression. As previously reported [9], Sox2 was expressed in most cases, both in the pre-anaplastic and in the anaplastic samples. However, by comparing expression levels in paired samples, at anaplastic progression we observed an increase in Sox2 expression in 40% of cases, a decrease in 30% cases, and no variations in the remaining 30% of cases.

### 3.5. Molecular Subgrouping and Prognostic Correlation of Progressive Anaplastic Meningiomas

Figure 3 shows the changes in molecular profile between pre-anaplastic and anaplastic samples. EGFRvIII gain was observed in three cases, while an increase in immunohistochemical Sox2 expression was observed in four cases. Interestingly, these changes were mutually exclusive, identifying three molecular subgroups. One group showed increased Sox2 expression at progression (and EGFRvIII stable), another group showed EGFRvIII gain (and stable or reduced Sox2) (*n* = 3), and a third group showed neither EGFRvIII gain nor Sox2 increase (*n* = 3). Noteworthily, the EGFRvIII group had a significantly prolonged survival compared with the Sox2 group (median OS 23 vs. 5 months; *p* = 0.0177, log-rank test; Figure 4). Cases with increased PD-L1 expression at progression had a significantly worse OS, as compared with those cases with stable PD-L1 expression (median OS 23 (7–26) vs. 2 (1.5–15) months; *p =* 0.0481, log-rank test) (Figure 5). No significant correlation was found between changes in EGFR expression and OS. Similarly, molecular profile, as assessed separately in pre-anaplastic or in anaplastic tumor, was not able to predict OS.

## 4. Discussion

In the present work, we analyzed the clinical and molecular features of anaplastic meningioma. Comparing paired pre-anaplastic and anaplastic meningioma samples, we also defined the molecular paths of tumor progression.

### 4.1. Clinical Data

Clinical results of our institutional series are in line with available literature, thus indicating the validity and generalizability of our results. In detail, the 2.7 years median OS of our series is in the reported range of 2.6–5.8 years [23,24]. The de novo phenotype of anaplastic meningioma correlated with more favorable OS, as previously reported [24]. Age and clinical status are well-established prognosticators [24]. As concerns the extent of resection, while GTR is a common goal in lower grade meningioma, its role in grade III meningioma management is still not clear [25,26]. In progressive meningioma, GTR can be very difficult to achieve due to previous treatments. Moreover, brain invasion is a non-negligible issue. Therefore, as an analogy to malignant brain tumors, the goal of surgery in malignant meningioma should be maximal safe resection.

### 4.2. Molecular Data

Mechanisms leading to progression in meningiomas are poorly understood. Accumulating literature data point to the role of molecular biology in predicting the prognosis of meningiomas across all grades [8,9,27,28,29]. Relatively few gene-level alterations, including *CDKN2A/B, TERT*, and *NDRG2* (N-myc downstream-regulated gene family member 2), have been strongly associated with the malignant progression of meningioma [30,31,32]. Activating *TERT* promoter mutations have been detected frequently in high-grade meningiomas and low-grade meningiomas that subsequently undergo malignant progression [33]. Nevertheless, *TERTp* mutation can occur independently of malignant progression in meningioma; it is most often present from the first tumor sample across recurring tumors, and in WHO grade III, it may represent a marker of an aggressive subset of tumors. Among other potential markers already investigated, defects in TGFβ and/or BMPs signaling and a decrease in the inhibitory regulation of TGFβ have been associated with meningioma progression [34]. A lot of recently published studies adopt advanced sequencing platforms, mainly NGS-based, to assess exome, transcriptome, and methylome of meningiomas. Nassiri et al. [8] and Sahm et al. [35] analyzed a series of 497 meningiomas for DNA methylation patterns, copy number aberrations, and mutations in genes known to be affected in meningioma. They succeeded in identifying six methylation classes with different patterns of cytogenetic aberrations, mutations, and histology, which showed up to be superior to the WHO classification in predicting clinical outcomes. Viaene et al., 2019 identified transcriptional signatures, distinguishing grade I tumors that will progress from those that will not, including GREM2, snoRNAs genes, and lower number of fusion transcripts [36]. However, although these techniques have become widely available, in the spirit of the molecular-oriented novel 2021 WHO classification of central nervous system tumors [1], NGS is not available for routine analysis in many centers, thus limiting the applied real clinical value of advanced molecular analyses. In the present work, we used simple and widely available techniques, such as immunohistochemistry and PCR, to draw the evolutional molecular profile of progressive anaplastic meningioma. We chose a series of molecular markers whose prognostic value had been already studied in other histotypes, and we evaluated their expression at different steps of progression of the same tumor in an institutional series of anaplastic meningioma patients. Herein, we report the main findings of our molecular study.

#### 4.2.1. Sox2 and EGFRvIII

Sox2 is a stemness marker with an established prognostic role in many tumors. In a previous study, we investigated its prognostic role in meningioma [9]. In that paper, we showed that Sox2 expression is a negative prognosticator of OS independently from tumor grade. Moreover, in progressive cases, we suggested that Sox2 expression may be an intrinsic feature of the tumor after its initial diagnosis. In the present study, we found a high percentage of Sox2-positive cases (80% in pre-anaplastic and 70% in anaplastic samples), thus confirming our previous findings. However, we observed an increase in Sox2 in four cases (Figure 3). Noteworthy, Sox2 increased expression was mutually exclusive with EGFRvIII gain, which was observed in 30% of cases (Figure 3). Cases with Sox2 increase had a significantly reduced OS compared to those with EGFRvIII gain (Figure 4). We speculate that a Sox2 increase reflects an epithelial-to-mesenchymal transition or, hierarchically, the acquisition of a stem-like phenotype [9,17]. Conversely, EGFRvIII gain at progression might reflect an epithelial phenotype or, hierarchically, a committed progenitor state [17], thus portending a reduced tumor aggressiveness. Literature background on Sox2 and EGFRvIII role in meningioma is scarce. Evidence of the EGFRvIII role in meningioma progression is conflicting and largely based on immunohistochemistry [37]. These suggestions thus need to be confirmed by further experiments with deeper mechanistic insights. Alternatively, the close relationship between Sox2 and EGFRvIII may reflect the embryonic origin of the arachnoid membrane, whereby meningioma expresses both epithelial and mesodermal antigens [38].

#### 4.2.2. PD-L1

Increased PD-L1 expression was a common finding in anaplastic meningioma progression and was associated with a worse prognosis (Figure 3 and Figure 5). We speculate that PD-L1 increase reflects an immune escape strategy by the progressing tumor, due to accumulating mutations [39]. Further studies, including mismatch repair and microsatellite instability analysis, are needed to confirm these hypotheses. The encouraging results from early clinical trials using checkpoint inhibitors, appear to confirm the role of immune escape mechanisms in meningioma progression [40].

#### 4.2.3. Other Molecular Markers

The reduced expression of EGFR during anaplastic progression, which was described by others [41], had no prognostic value. VEGF was mostly overexpressed both before and after progression, and it is not related with prognosis. VEGF overexpression suggests a potential effectiveness of a therapy with angiogenesis inhibitors, however, evidence is still scarce [42]. *MGMT* promoter was mostly un-methylated, reflecting the poor efficacy of alkylating agents in these tumors [43,44]. Finally, and surprisingly, *TERT* promoter status was wild-type in all cases. A discrepancy between telomerase hyperactivity and absence of *TERT* promoter mutation had already been described: Goutagny et al. [33] found *TERT* promoter mutations in just 15% of grade III meningiomas and telomerase hyperactivity in about 95% of cases. Telomerase hyperactivity in malignant meningioma was also reported by our group [45]. Further studies on the mechanisms regulating *TERT* hyperactivity in meningiomas are thus needed.

### 4.3. Strengths and Limitations of the Present Study

The main limitations of the present study are its retrospective nature and the limited patient numerosity. Moreover, analyses were performed only on progressive and not on de novo meningiomas. However, patients were homogeneously treated and followed up at a single nationwide reference neuro-oncologic center; thus, it was reassuring in terms of quality and reliability of data.

## 5. Conclusions

Grade III meningiomas are malignant tumors endowed with a grim prognosis. Progressive cases portend a particularly unfavorable survival. The role of longer clinical history and previous treatments on one side and genetics on the other side in contributing to the poor prognosis still have to be explored in depth. In the present study, we identified two subgroups, a stem-like/mesenchymal subgroup and an epithelial one, driven, respectively, by Sox2 and EGFRvIII, with different prognosis. Moreover, we found that progression is associated with immunosuppression, as shown by increased PD-L1 expression, leading to worsened OS. These results, in addition to their prognostic role, could pave the way for future studies with personalized treatments aimed at targeting actionable molecular drivers. The improvement in the understanding of biological bases of malignant progression in meningioma may lead to better patient risk stratification and guide clinical decisions in the era of precision medicine.

## Figures and Tables

**Figure 1 jpm-13-00206-f001:**
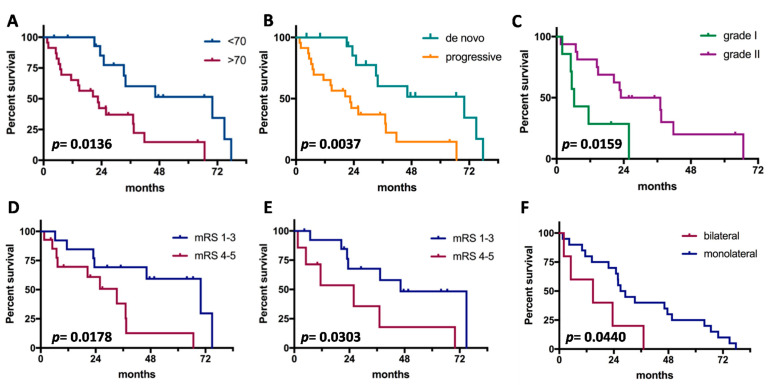
Prognostic factors for survival in anaplastic meningiomas. (**A**), Age; (**B**), tumor subtype (de novo vs. progressive); (**C**), grade of origin (limited to progressive cases); (**D**), preoperative clinical status; (**E**), postoperative clinical status; (**F**), laterality of tumor.

**Figure 2 jpm-13-00206-f002:**
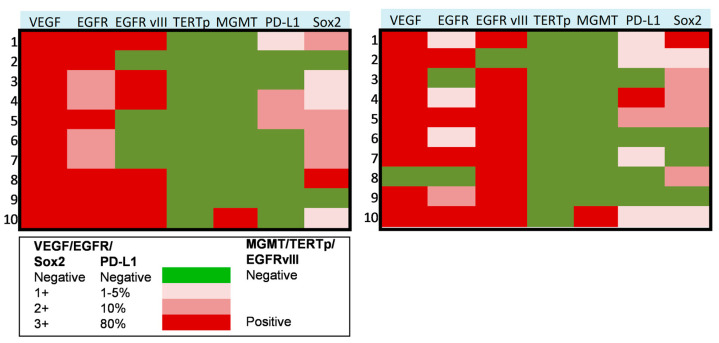
Heat maps showing the molecular profile of progressive anaplastic meningiomas at first diagnosis (**left**) and at anaplastic progression (**right**). X axis: molecular markers expression, as reported in the legend. Y axis: patients of the series.

**Figure 3 jpm-13-00206-f003:**
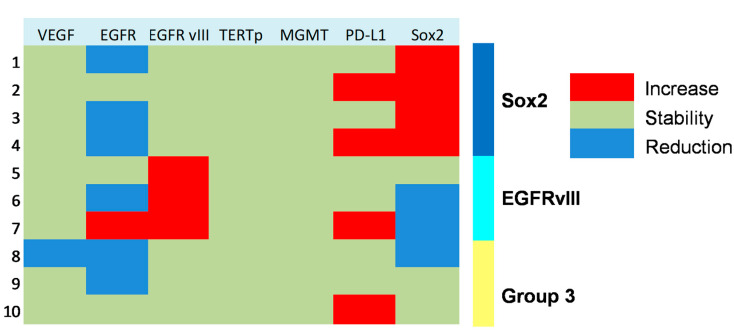
Heat map showing molecular subgrouping of progressive anaplastic meningiomas based on changes of expression of key markers at tumor progression.

**Figure 4 jpm-13-00206-f004:**
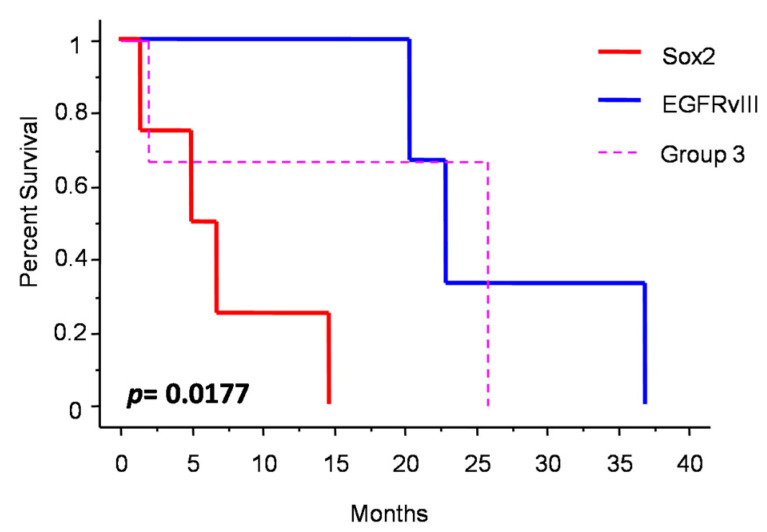
Patients survival after diagnosis of anaplastic meningioma depending on molecular subgrouping.

**Figure 5 jpm-13-00206-f005:**
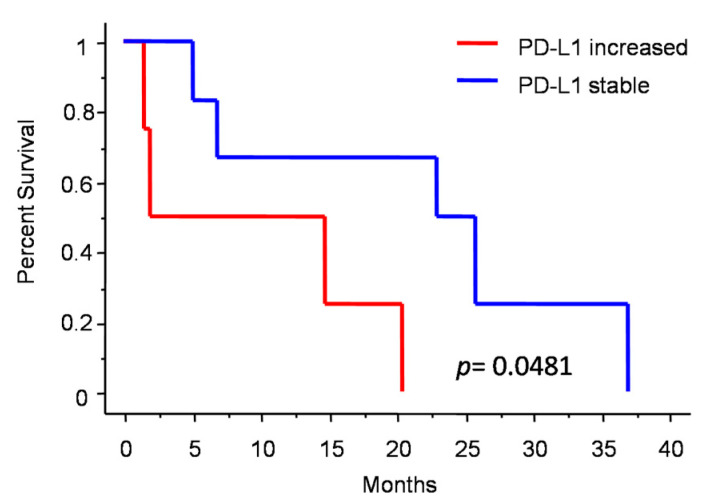
Patients survival after diagnosis of anaplastic meningioma depending on PD-L1 change.

**Table 1 jpm-13-00206-t001:** Baseline characteristics of patients.

Parameter	De Novo	Progressive	*p*
*N*	17 (41.5%)	24 (58.5%)	NA
Male sex	41.2%	62.5%	0.2159 *
Age at diagnosis (years)	68.8 (19.2–84.6)	63.7 (50.3–83.4)	0.5027 ^#^
Tumor location			
Non-skull base	88.2%	54.2%	0.0623 **
Skull base	5.9%	33.3%	
Intraventricular	5.9%	12.5%	
Preop mRS ≤ 3	88.2%	79.2%	0.6786 *
Postop mRS ≤ 3	81.8%	58.3%	0.3707 *

*, Fisher’s exact test; **, Chi-square test; ^#^, Mann–Whitney U test. NA, not applicable.

**Table 2 jpm-13-00206-t002:** Surgery and follow-up.

Parameter	De Novo	Progressive	*p **
Single surgery	64.7%	62.5%	>0.99
GTR	100%	64.7%	0.0237
Surgical complication	21.4%	47.1%	0.2580
Adjuvant radiotherapy	90.9%	84.6%	>0.99
Chemotherapy	0	20.8%	0.0650
Extracranial metastasis	0	8.3%	0.5024

*, Fisher’s exact test. GTR, gross total resection. bevacizumab. Extracranial metastases occurred in two progressive cases and involved kidney, liver, and lung.

**Table 3 jpm-13-00206-t003:** Multivariate analysis for OS.

Parameter	Hazard Ratio	Confidence Interval	*p*
De novo vs. progressive	0.149	0.020–1.123	0.0647
mRS ≤3 vs. >3	0.484	0.091–2.584	0.3961
Age ≤70 vs. >70	6.479	0.815–51.488	0.0772
Bilateral vs. monolateral	3.154	0.313–31.797	0.3299
GTR vs. STR	4.591	0.3–70.246	0.2735

GTR, gross total resection; mRS, modified Rankin scale; STR, subtotal resection.

## Data Availability

Source data are available from the corresponding author upon reasonable request.

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
