# Peer review of "Paths of Evolution of Progressive Anaplastic Meningiomas: A Clinical and Molecular Pathology Study"

_jpm, 2023, doi:10.3390/jpm13020206_

Round 1
Reviewer 1 Report
Review Report for the Manuscript “Paths of evolution of progressive anaplastic meningiomas. A clinical and molecular pathology study”
Rating the Manuscript
Originality/Novelty: Is the question original and well defined? Do the results provide an advance in current knowledge?
Yes, in the manuscript the authors focus and study anaplastic meningiomas using techniques including immunohistochemistry and PCR, to assess the molecular paths of evolution of progressive anaplastic meningiomas.
Significance: Are the results interpreted appropriately? Are they significant? Are all conclusions justified and supported by the results? Are hypotheses and speculations carefully identified as such?
Yes, the results are interpreted well.
Quality of Presentation: Is the article written in an appropriate way? Are the data and analyses presented appropriately? Are the highest standards for presentation of the results used?
Yes, the article is written well. Data representation could be improved. Specially the figure captions could be more informative.
Scientific Soundness: is the study correctly designed and technically sound? Are the analyses performed with the highest technical standards? Are the data robust enough to draw the conclusions? Are the methods, tools, software, and reagents described with sufficient details to allow another researcher to reproduce the results?
Yes, the data is robust enough to draw conclusions and the methods, tools and methods used in the data analysis are explained properly.
Interest to the Readers: Are the conclusions interesting for the readership of the Journal? Will the paper attract a wide readership, or be of interest only to a limited number of people? (Please see the Aims and Scope of the journal)
Yes, this would be a great article for the researchers in the cancer research field.
Overall Merit: Is there an overall benefit to publishing this work? Does the work provide an advance towards the current knowledge? Do the authors have addressed an important longstanding question with smart experiments?
Yes. This study provides an advancement to the current knowledge.
English Level: Is the English language appropriate and understandable?
Yes, English language in the manuscript is appropriate and understandable.
Overall Recommendation: Accept after Minor Revisions
Given below are the comments for each section of the manuscript.
Abstract
The abstract is written and summarizes the content of the manuscript.
Line 20: “VEGF, EGFR, EGFRvIII, PD-L1, and Sox2 expression, MGMT methylation status and TERT promoter mutation were assessed in paired meningioma samples collected from the same patient before and after progression using immunohistochemistry and PCR.”
It’s better if the authors could define the terms like VEGF, EGFR, MGMT and TERT when these first appears in the manuscript.
Introduction
Introduction is well written.
In my opinion it’s great if the authors could discuss about the statistics on Meningiomas and survival rates. These facts will help to emphasize the importance of this study.
Line 38: “Anaplastic meningiomas represent only 5% of all meningioma tumors but are overt malignancies with a dismal prognosis.”
It’s better if the authors can discuss about Anaplastic meningiomas in detail.
Line 51: “The present work focuses on anaplastic meningiomas. More specifically, we used widely available techniques, like immunohistochemistry and PCR, to assess the molecular paths of evolution of progressive anaplastic meningiomas.”
Briefly mention what molecular paths studied in this manuscript.
Materials and Methods:
2.2. Molecular analysis
Line 72: “We evaluated the expression of VEGF, EGFR, EGFRvIII, PD-L1, Sox 2, MGMT methylation status, and TERT promoter mutation.”
Authors need to explain why they selected these genes for expression analysis. They include this in the discussion section.
Line 74: “VEGF expression was evaluated using immunohistochemistry, EG-FRvIII expression using RT-PCR and MGMT promoter methylation using methylation-specific PCR. Sox2 expression was evaluated using immunohistochemistry as already described.”
Why different techniques are used to analyze the expression of these genes?
Line 77: “A sample was scored as positive where ≥25% cells showed nuclear or nuclear-cytoplasmic expression of Sox2.”
Is it a standard method to consider a sample as positive sample if >25% cells are positive for nuclear or nuclear-cytoplasmic expression of Sox2? If so, authors need to include a reference for this.
Line 80: “EGFR expression was scored as 0, 1+, 2+ or 3+ in case of positivity of >25%, 25-50%, 50-75% or >75% of tumor cells, respectively (Supplementary Fig. S1).”
Is it a standard way to categorize the stage based on these values? if so, authors need to include reference here.
Line 86: “The annealing temperature was 62°C. The 230-bp amplified product was purified by adding 2 μl of ExoSap (USB Corporation, Cleveland) at 37°C for 15’, and at 80°C for 15’.”
Is this 15 min?
Results
3.5. Molecular subgrouping and prognostic correlation of progressive anaplastic meningiomas
Line 172: “Noteworthy, the EGFRvIII group had a significantly prolonged survival compared with the Sox2 group (median OS 23 vs 5 months; p=0.0177, log-rank test; Figure 4).”
Please mention the CI whenever you state the p value.
3.2 Figures and Tables
Representation of data and tables could be improved.
The figure captions need to be more informative.
Figure 2 and 3: In the figure caption please discuss what the axes represent in the heat maps.
Discussion
Line 206: Molecular data
This section looks more like a part of an introduction. Authors could reorganize or reword this paragraph.
Other than the markers discussed in this manuscript, are there any other markers that have been discussed in literature?
References:
Some of the references are more than 10 years old. It they don’t contain important information authors could replace these with new references.
References: 4,5,10,12,20,23,25,26 and 28
Reviewer 2 Report
Review of manuscript ”Paths of evolution of progressive anaplastic meningiomas. A clinical and molecular pathology study”. The authors have studied molecular markers in patients with progressive anaplastic meningiomas and analyzed these markers in a subset of 10 patients at first surgery (pre-anaplastic) and at anaplastic progression. The patient cohort is rather small, but their findings are interesting as we know the histological assessment often not is correlated to biological behavior and identifying molecular markers is highly relevant in this group. I have some comments.
Abstract
I suggest adding how many cases you included.
Introduction
The second sentence seems incomplete, please replace XX with which century you want to write. Also, should it be Cushing et Eisenhardt?
Results
The last sentence under “Follow-up” lacks ending.
Methods
Any reason why the molecular analysis wasn’t performed in all patients for comparison with de novo cases? I see this is mentioned under Limitations.
Results
The figures are now under Discussion, I suggest moving them to Results.
